# Internal Carotid Artery Injury in Transsphenoidal Surgery: Tenets for Its Avoidance and Refit—A Clinical Study

**DOI:** 10.3390/brainsci11010099

**Published:** 2021-01-13

**Authors:** Dmitry Usachev, Oleg Sharipov, Ashraf Abdali, Sergei Yakovlev, Vasiliy Lukshin, Maksim Kutin, Dmitry Fomichev, Pavel Dorokhov, Evgeny Bukharin, Alexey Shkarubo, Ilya Chernov, Andrey Panteleyev, Kaan Yağmurlu, Bipin Chaurasia, Pavel Kalinin

**Affiliations:** 1N.N. Burdenko National Medical Research Center of Neurosurgery 4-ya Tverskaya-Yamskaya 16 st., 125047 Moscow, Russia; Dousachev@nsi.ru (D.U.); osharipov@nsi.ru (O.S.); sysb@nsi.ru (S.Y.); wlukshin@nsi.ru (V.L.); Kutin@nsi.ru (M.K.); DFomicev@nsi.ru (D.F.); PDorochov@nsi.ru (P.D.); GBuharin@nsi.ru (E.B.); AShkarubo@nsi.ru (A.S.); IChernov@nsi.ru (I.C.); pkalinin@nsi.ru (P.K.); 2Department of Neurological Diseases and Neurosurgery, Peoples’ Friendship University of Russia (RYDN University), Miklukho-Maklaya Street, 117198 Moscow, Russia; 3Department of Spine Pathology, N.N. Priorov Central Institute of Traumatology and Orthopedics, 125047 Moscow, Russia; apanteleyev@gmail.com; 4Departments of Neurosurgery and Neuroscience, University of Virginia, Charlottesville, VA 22903, USA; ky7zb@virginia.edu; 5Department of Neurosurgery, Terai Hospital and Research Hospital, Birgunj 44400, Nepal; trozexa@gmail.com

**Keywords:** endoscopic transsphenoidal surgery, internal carotid artery injury, high-flow bypass, reconstruction, stent graft

## Abstract

One of the most serious/potentially fatal complications of transsphenoidal surgery (TSS) is internal carotid artery (ICA) injury. Of 6230 patients who underwent TSS, ICA injury occurred in 8 (0.12%). The etiology, possible treatment options, and avoidance of ICA injury were analyzed. ICA injury occurred at two different stages: (1) during the exposure of the sella floor and dural incision over the sella and cavernous sinus and (2) during the resection of the cavernous sinus extension of the tumor. The angiographic collateral blood supply was categorized as good, sufficient, and nonsufficient to help with the decision making for repairing the injury. ICA occlusion with a balloon was performed at the injury site in two cases, microcoils in two patients, microcoils plus a single barrel extra-intracranial high-flow bypass in one case, stent grafting in one case, and no intervention in two cases. The risk of ICA injury diminishes with better preoperative preparation, intraoperative navigation, and ultrasound dopplerography. Reconstructive surgery for closing the defect and restoring the blood flow to the artery should be assessed depending on the site of the injury and the anatomical features of the ICA.

## 1. Introduction

Analysis of the surgical treatment results of pituitary tumors via endoscopic transsphenoidal access demonstrates the effectiveness and safety of this approach and minimal postoperative complications and mortality. However, damage to the internal carotid artery remains one of the most serious and potentially fatal complications of transsphenoidal surgery, which, according to various authors, is observed in up to 3.8% of cases [1,2,3,4,5,6]. We describe our clinical experience with internal carotid artery injury during transsphenoidal surgery based on its causes, treatment, and prevention strategies and compare them to other experiences in the literature.

## 2. Materials and Methods

Between January 2005 and March 2020, our neurosurgical center conducted more than 6230 endoscopic transsphenoidal surgeries for removing chiasmosellar and pituitary region tumors. Injury to the internal carotid artery was registered in 8 of these patients, which amounted to 0.12% of all cases. The study group also included a case of pituitary adenoma, in which damage to the internal carotid artery occurred during transsphenoidal surgery performed in a different clinic. This complication was also treated in our center. The age of the patients ranged from 24 to 66 years (median 43.5 years). Three of the patients were male and five were female. Six cases were operated on for pituitary adenomas, one case for a chordoma, and one case for a trigeminal neuroma.

In all cases, damage to the cavernous segment of the internal carotid artery (ICA) was accompanied by massive bleeding. Blood from a defect in the ICA wall filled the surgical field instantaneously, and we had to use double suction to better visualize the surgical area. In one case, the endoscope was replaced with an operating microscope to control the bleeding. Hemostasis involved tight packing of the site of bleeding using gauze and all other feasible hemostatic materials.

After stopping the bleeding in these patients, angiography was performed to determine the defect’s location in the arterial wall and evaluate the possible collateral circulation to select further surgical strategies in consultation with endovascular neurosurgeons. An intraoperative video of case 7 in Table 1 demonstrates ICA injury during endonasal endoscopic opening of the cavernous sinus’s medial wall and tight-packing hemostatic material with successful hemostasis. The video also shows the occurrence of a pseudoaneurysm at the injury site and its treatment with endovascular microcoils during postoperative angiography (Appendix A).

## 3. Results

Our treatment modalities for ICA injury are presented in Table 1.

The causes of ICA damage during TSS can be divided into two major groups:

(1)Injury during exposure: (a) deviation/displacement of the access trajectory relative to the midline occurred in two cases and (b) durotomy at the sella floor or in the cavernous sinus region occurred in three cases.(2)Injury during tumor removal: Manipulations in the cavernous sinus occurred in three cases. Cerebral DSA was performed in all patients after fixing the ICA injury immediately after surgery. In six cases, a false aneurysm was revealed; in one case, a defect in the ICA wall was not detected. In one patient, ICA occlusion was observed, which was associated with either ICA spasm or excessive tamponade of the removed tumor’s cavity. In one case (no. 4), immediately after carotid artery injury, no changes were detected during angiography; however, during repeat angiography on the eighth day after surgery, a false aneurysm was detected.

Besides evaluating the arterial injury, all patients underwent collateral circulation assessment to determine the further treatment strategy. During the collateral circulation assessment, angiography was used to determine the presence and degree of anterior and posterior communicating artery development and filling of the anterior and middle cerebral arteries on the side of the ICA injury (Table 2). Three degrees of collateral circulation were identified: (1) good circulation—the presence of large anterior and posterior communicating arteries and complete filling of the distal branches of the anterior and middle cerebral arteries with the contrast, on the side of the ICA injury through the contralateral ICA; (2) sufficient circulation—the presence of hypoplasia or the absence of it of the communicating arteries with a little filling of the distal branches of the middle or anterior cerebral artery with the contrast; and (3) nonsufficient circulation—the absence of collateral circulation in the cerebral arterial circle. The possibility of reconstructive surgery for closing the defect and restoring the blood flow was assessed depending on the injury’s location and the anatomical features of the ICA. When reconstruction was not possible, after determining the potential collateral blood flow, defect closure by occlusion of the ICA lumen with consecutive revascularization of arterial circulation in the ICA blood supply zone (Figure 1) was elected. In the case of nonsufficient collateral circulation, before ICA occlusion, flow-substituting surgical interventions were performed in the form of an overlapping high-flow bypass capable of compensating for the cerebral blood flow by more than 100 mL/min.

When sufficient collateral circulation was observed, ICA occlusion was performed as the first stage, followed by evaluating the perfusion deficit. If a deficit was confirmed, flow-supplementing surgical intervention in the form of a single or double extra-intracranial bypass was performed as the second stage.

In cases of adequate collateral blood flow, endovascular occlusion at the ICA injury site using balloons in two cases and microcoils in three cases was performed (Figure 2).

Hemiparesis occurred in one case with sufficient collateral circulation after occlusion of the injured ICA. After endovascular intervention, DWI MRI and MSCT-perfusion images showed a decrease in the cerebral blood flow and ischemic zone in the middle cerebral artery (MCA) territory on the side of the occluded ICA (Figure 3). Therefore, the patient underwent single-barrel extra-intracranial anastomosis between the superficial temporal artery and the middle cerebral artery on the left side.

In the case of nonsufficient collateral circulation, the approach to closing the defect required maintaining the blood flow in the basin of the injured artery (reconstructive surgery) or replacing the blood flow (bypass) in the MCA basin on the ipsilateral side: in one case, the patient underwent a stent-graft procedure (Figure 4), and one patient underwent a high-flow bypass followed by ICA occlusion using microcoils (Figure 5).

In one case (clinical case 1) with angiographic signs of ICA occlusion and adequate collateral circulation, endovascular treatment was not performed. In another case (clinical case 6), which involved defect closure with a muscle fragment, endovascular surgery was also not required (Table 2).

After closing the defect (balloon ICA occlusion, stent graft, muscle fragment tamponade) of the internal carotid artery, no symptoms or complications were observed in three cases (37.5%). Neurological deterioration was observed in three cases (37.5%): one patient presented with visual field defects and transient hemihypesthesia, one patient developed hemiparesis, and one patient developed transient abducent nerve paresis. The mortality rate was 25% (*n* = 2); in both cases, the cause of death was the development of total cerebral hemisphere ischemia on the injured ICA side.

## 4. Discussion

Carotid artery injury during transsphenoidal surgery is primarily associated with losing the surgical access trajectory with deviation from the midline and manipulations in the cavernous sinus. Preoperative planning of transsphenoidal surgical interventions is crucial to prevent iatrogenic carotid artery injury. It is essential to thoroughly assess the anatomy of the chiasmosellar region based on preoperative MRI data, determine the size of the sphenoid sinus and the degree of its pneumatization, and adequately identify medial displacements of the ICA cavernous segment, if present [7]. It is also essential to bear in mind that the septum (or septae) of a sphenoid sinus is not a reliable landmark for determining the midline, since it can often be displaced to one side. MRI visualization can help determine the exact location of the cavernous segment of the ICA based on the sella floor. An increased risk of injury to the ICA during transsphenoidal surgery exists in patients with acromegaly due to changes in the anatomy of the nasal cavity and sphenoid sinus and decreased distance between bilateral cavernous segments of the ICA [8]. Intraoperative navigation is essential for real-time visualization and identification of anatomical structures in the sellar region during repeat surgeries, in cases with underdevelopment of the sphenoid sinus in children, when the prominent anatomical landmarks of the midline and the location of the carotid artery cannot be identified. Intraoperative ultrasound dopplerography also helps identify the ICA on both sides, especially for removing parasellar neoplasms and sellar neoplasms extending into the cavernous sinus, middle cranial fossa, trigeminal cavity, or pterygopalatine fossa. After introducing this technique into our daily practice, no iatrogenic ICA injury cases were observed [9]. The surgical outcome and the number and severity of intraoperative complications directly correlate with the number of transsphenoidal surgeons’ experiences [5]. ICA damage was observed at a rate of 1.4% in a group of patients operated on by surgeons with little experience in transsphenoidal surgery (fewer than 200 surgeries) and 0.4% in a group of patients operated on by more experienced surgeons (more than 500 surgeries) [10].

The endoscopic transsphenoidal approach is the method of choice for surgical treatment of the majority of pituitary adenomas. It is useful for removing other chiasmosellar region tumors (craniopharyngiomas, meningiomas, chordomas, etc.). The risk of ICA injury, associated with a relatively high number of neurological complications and mortality, remains a relatively rare transsphenoidal surgery complication. Capabianca et al. reported one ICA injury in 146 patients who underwent transsphenoidal surgery (0.68%) [4]. Tabaee et al. conducted a review of the literature (nine articles by various authors, including a total of 821 endoscopic transsphenoidal surgeries), identifying two deaths due to ICA injury, which was 0.24% of all reported cases [11]. Kassam et al. reported two ICA injury cases in 800 endoscopic surgeries performed (0.25%) [5]. The ICA iatrogenic injury rate in 6000 patients who underwent TSS at our institute was 0.12%.

Injury to the internal carotid artery during transsphenoidal surgery leads to a complex and urgent surgical problem/complication, which requires a combined treatment approach involving several surgical teams. The aim of emergent endovascular surgical treatment is to close the defect in the artery, and in cases with the absence of normal collateral circulation, it involves endovascular intervention and/or bypass surgery (flow-supplementing interventions) (see Figure 1).

In the event of profuse bleeding, precise coordination between the surgeon and the assistant is paramount. To stop the bleeding temporarily, pressure can be applied to the common carotid artery in the neck. When the ICA is injured, the intensity of bleeding is massive. Therefore, single suction is not sufficient for adequate surgical field visualization; consequently, it is advisable to simultaneously use double surgical suction pumps. The surgeon’s actions should be aimed at promptly packing the site of bleeding with any available absorbent material for local hemostasis. We can also use a muscle patch to pack the injured ICA and absorbent material as tamponade in the sphenoid sinus. In our study, ICA bleeding was successfully stopped using muscle fragment tamponade in one case (clinical case no. 6), which was confirmed by the absence of extravasation of the contrast material into extravascular locations during cerebral angiography. This strategy, however, is not always practical. According to Gardner et al. [12], in seven patients with ICA injury during transsphenoidal surgery (0.3% of the total patients in the series), bleeding from the injured vessel was successfully stopped by tamponade of the sinus cavity with a muscle fragment in only one case.

In some cases, it is possible to coagulate small arterial defects using bipolar coagulation [5,12] or perform ICA trapping at the injury site. Kassam et al. [5] described a case of successful clipping via the endoscopic endonasal route for an injured ICA distal and proximal to the defect. The procedure was associated with significant technical difficulties and was accompanied by ischemic complications.

It is necessary to focus on the anatomical features of the artery and the variability of collateral circulation to decide what kind of endovascular procedure to use (see Figure 1). Endovascular vessel occlusion at the site of injury or stent-graft installation has been proposed as the most effective method of treating iatrogenic injury to the cavernous segment of the ICA [13,14]. It is not always possible to immediately identify the defect after ICA injury via angiography. In these cases, repeated angiographic studies must be performed on postoperative day 7 or 8 [13].

In the presence of compensated (good) collateral blood circulation, occlusion of the internal carotid artery at the injury site is recommended. A latex balloon catheter was used for this purpose in cases 2 and 3. However, given the possibility of vessel recanalization due to a decrease in the balloon volume and the increased risk of enlargement of the existing defect in the vessel wall after balloon inflation, this technique is no longer used. An alternative to this technique is endovascular occlusion of the ICA with microcoils, used in cases 5 and 7.

In cases with sufficient collateral circulation, ICA occlusion should be considered. However, given the increased risk of ischemic complications, patients should undergo perfusion studies in the postoperative period to assess the need for further flow-supplementing surgery, i.e., single or double extra-intracranial bypass.

It is significantly more challenging to close the ICA defect under nonsufficient collateral circulation conditions, leading to ischemic stroke after occlusion of the injured vessel. A stent graft can be used to close the defect while maintaining the lumen of the artery. This technique, however, is associated with several limitations: The stent can be placed over restricted sites of the ICA, in locations where no major collateral efferent arteries to the ICA exist; it cannot be placed at the bent standard anatomical curves of the ICA in the cavernous sinuses; and a stent graft is associated with a higher risk of thrombosis [7,13]. A high-flow bypass can compensate for cerebral blood flow in patients with nonsufficient collateral circulation to prevent ischemic complications. This approach was successfully used in the surgical treatment of giant ICA aneurysms [15,16,17].

A critical factor in choosing bypass surgery is the preoperative assessment of potential collateral circulation. A low-flow bypass between the branches of the superficial temporal and middle cerebral arteries may not fully compensate for the deficiency of cerebral circulation in the occluded ICA basin, leading to severe ischemic complications [1,8,18]. Hence, high-flow bypass surgery is indicated. In a study by Rangel-Castilla et al. [19] of eight patients with ICA injury, large-caliber extracranial-to-intracranial bypasses with proximal MCA branches were performed on the side of occlusion without severe ischemic complications.

One of the variations in the course of acute cerebrovascular accident development is the stroke in progression, which is characterized by a progressive onset of/increase in symptoms within hours and even days [20]. This type of acute cerebrovascular accident was observed in two cases with lethal outcomes despite angiographic signs of adequate circulation in the cerebral arterial circle. For early detection of such complications after ICA injury, transcranial doppler ultrasonography of the MCA on the side of ICA occlusion is recommended. Cerebral perfusion assessment in the early postoperative period to identify ischemic events and determine indications for emergency flow-supplementing procedures (bypass) is also highly advised.

In the postoperative period, adequate intensive care, including controlled arterial hypertension (intravenous infusion of sympathomimetic and volemic control) with a targeted systolic blood pressure of 160–180 mm Hg, sedation, and monitoring of the rheological properties of blood, is of particular importance.

## 5. Conclusions

Meticulous neuroimaging data analysis and the surgeon’s experience can significantly reduce the risk of iatrogenic ICA injury during transsphenoidal approach. When removing the pituitary gland and parasellar region tumors using endoscopic transsphenoidal access, correct identification of the midline is of paramount importance. To prevent complications, it is advisable to use intraoperative navigation and ultrasound dopplerography to visualize the ICA during procedures performed in the parasellar region. Dealing with ICA injury requires detailed angiographic examination and a combined multidisciplinary treatment approach.

## Figures and Tables

**Figure 1 brainsci-11-00099-f001:**
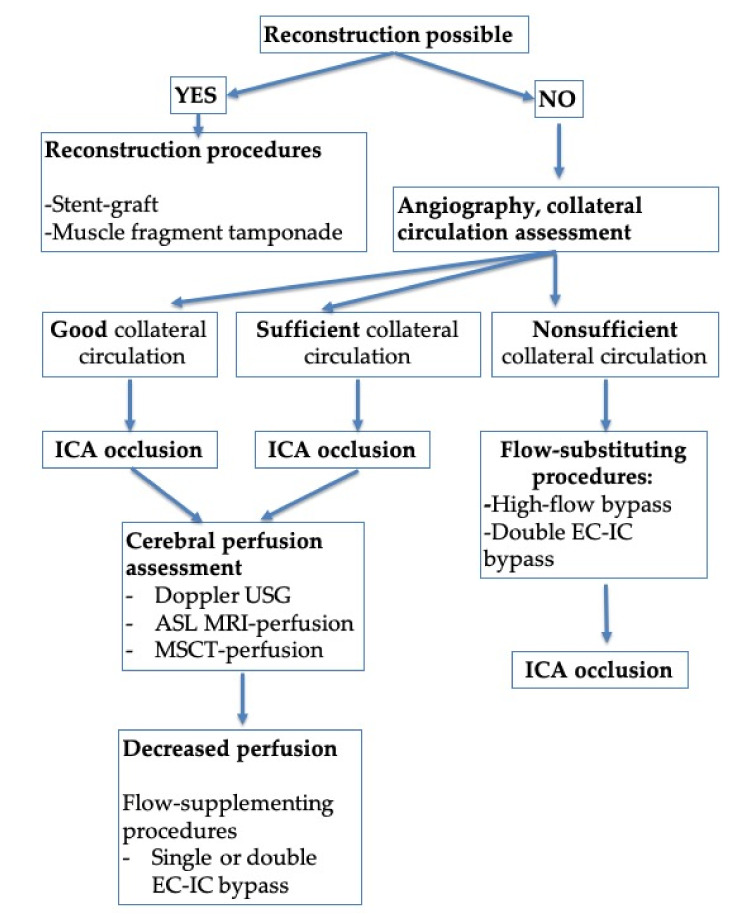
Treatment algorithm for patients with ICA injury (depending on anatomical features of the artery and potential for surgical reconstruction). Abbreviations: ASL MRI, arterial spin labeling magnetic resonance imaging; EC, extracranial; IC, intracranial; ICA, internal carotid artery; MSCT, multislice computed tomography; USG, ultrasound.

**Figure 2 brainsci-11-00099-f002:**
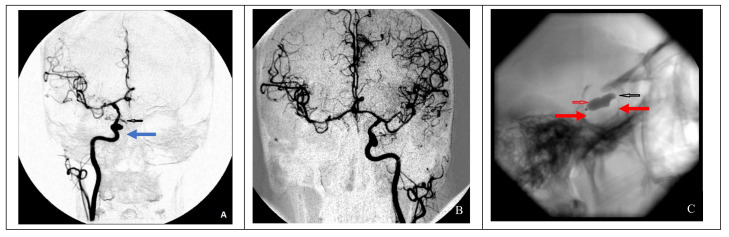
Endovascular surgical treatment in cases with good collateral circulation (clinical case 2). (**A**) A false aneurysm in the area of the cavernous segment of the right ICA (indicated by the blue arrow), (**B**) cerebral angiography of the left ICA demonstrating good collateral blood flow to the contralateral side, and (**C**) occlusion of a defect in the right ICA using two balloons (indicated by red arrows).

**Figure 3 brainsci-11-00099-f003:**
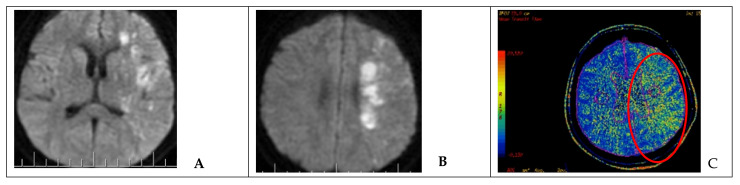
Neuroimaging in a patient after endovascular occlusion of the ICA (clinical case 5). (**A**,**B**) MRI of the brain (DWI); an area of acute ischemic injury in the left hemisphere (hyperintense area) can be visualized. (**C**) MSCT-perfusion study demonstrating signs of cerebral circulation insufficiency in the form of prolonged blood transit time in the basin of the left MCA (red circled area).

**Figure 4 brainsci-11-00099-f004:**
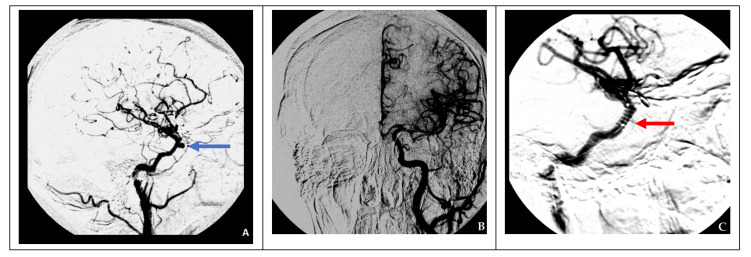
Endovascular treatment in a case of nonsufficient circulation (clinical case 4). (**A**) Formation of a false aneurysm at the level of the cavernous segment of the right ICA (blue arrow), (**B**) cerebral angiography of the left ICA indicating nonsufficient collateral circulation on the right side, and (**C**) installation of a stent graft (red arrow) at the site of the right ICA injury.

**Figure 5 brainsci-11-00099-f005:**
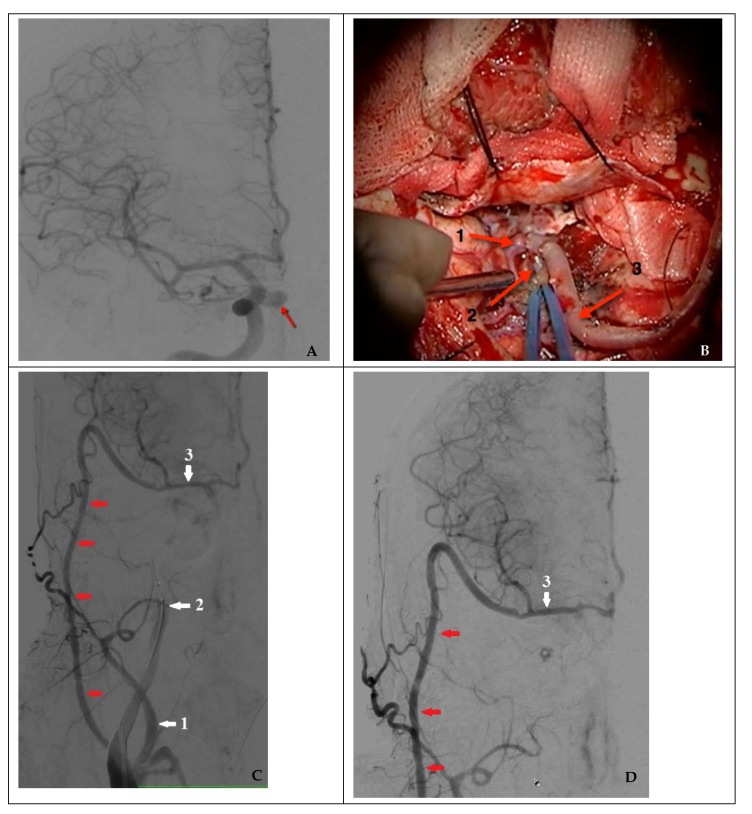
Combined treatment in a case of nonsufficient circulation (clinical case 8). (**A**) Angiography of the right ICA (the arrow indicates a false aneurysm), (**B**) view of a high-flow bypass with right MCA M2 segment bifurcation (1: frontal branch of M2; 2: distal sections of the temporal branch of M2; and 3: radial artery as the graft), (**C**) functioning high-flow bypass (indicated by red arrows) during a temporary test ICA occlusion, and (**D**) control angiography after endovascular occlusion of a false aneurysm and the injured right ICA (1: external carotid artery; 2: internal carotid artery occluded by a balloon; and 3: middle cerebral artery).

**Table 1 brainsci-11-00099-t001:** Data of patients with ICA injury, treatment options, and outcomes.

Case	Age/Sex	Tumor	ICA Damage Cause	Angiography	Treatment	Outcome
1	55 yo/F	Pituitary adenoma	Access deviation relative to the midline	ICA occlusion	Conservative therapy	Lethal
2	61 yo/M	Pituitary adenoma	Durotomy at the bottom of the sella turcica (with medial ICA displacement)	False aneurysm	Balloon ICA occlusion at the damage site	Lethal
3	40 yo/F	Pituitary adenoma	Access deviation relative to the midline	False aneurysm	Balloon ICA occlusion at the damage site	No complications
4	46 yo/M	Pituitary adenoma	Manipulation inside the cavernous sinus	False aneurysm	Installation of a stent-graft at the damage site	No complications
5	66 yo/F	Pituitary adenoma	Manipulation inside the cavernous sinus	False aneurysm	Occlusion of ICA with micro coils at the damage site + single extra-intracranial anastomosis between the superficial temporal artery and the middle cerebral artery	Hemiparesis
6	24 yo/F	Chordoma	Durotomy in the cavernous sinus area	ICA defect not detected	Intraoperative muscle tamponade of the defect	No complications
7	25 yo/F	Trigeminal neuroma	Durotomy in the cavernous sinus area	False aneurysm	Occlusion of microcoils at the damage site	Transient paresis of CN VI
8	41 yo/M *	Pituitary adenoma	Manipulation in the region of the cavernous sinus	False aneurysm	High-flow bypass + occlusion of the ICA with microcoils at the damage site	Visual field defect, transient hemihypesthesia

* The patient underwent initial surgery at a different clinic. ICA, internal carotid artery; CN, cranial nerve.

**Table 2 brainsci-11-00099-t002:** ICA defect closure approach depending on the degree of collateral circulation.

Case	Degree of Collateral Circulation	Defect Closure Approach
1	Good	Conservative therapy
2	Good	ICA occlusion with a balloon at the site of injury
3	Good	ICA occlusion with a balloon at the site of injury
4	Nonsufficient	Stent-graft
5	Sufficient	ICA occlusion with microcoils at the site of injury + single extra-intracranial bypass (flow-supplementing surgery)
6	Good	No endovascular treatment needed
7	Good	ICA occlusion with microcoils at the site of injury
8	Nonsufficient	High-flow bypass (flow-substituting surgery) + ICA occlusion with microcoils at the site of injury.

## Data Availability

Copyright is retained by the authors. This article is licensed under the open access Creative Commons CC BY 4.0 license.

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
