# Peer review of "Internal Carotid Artery Injury in Transsphenoidal Surgery: Tenets for Its Avoidance and Refit—A Clinical Study"

_brainsci, 2021, doi:10.3390/brainsci11010099_

Round 1

Reviewer 1 Report

The authors have reported the eight patients (0.12%) out of 6230 patients with ICA injury during endoscopic transsphenoidal surgery.   This case series provides transsphenoidal/pituitary surgeons with useful information about crisis control of intraoperative ICA injury.

To increase value of this article, the authors should demonstrate intraoperative view pictures immediately before ICA injuries. Pictures of the operative field immediately before an ICA injury are valuable for transsphenoidal/pituitary surgeons to avoid this catastrophic incident.

Furthermore, operative video digests from immediately before ICA injury to completion of hemostasis are extremely valuable. The authors are expected to provide the operative video digests. 

Author Response

Reviewer 1

Reviewer:  The authors have reported the eight patients (0.12%) out of 6230 patients with ICA injury during endoscopic transsphenoidal surgery.   This case series provides transsphenoidal/pituitary surgeons with useful information about crisis control of intraoperative ICA injury.

Respond#: We appreciate the reviewer for his/her careful review of our paper. We believe that our manuscript has been improved significantly.  

Reviewer:  To increase value of this article, the authors should demonstrate intraoperative view pictures immediately before ICA injuries. Pictures of the operative field immediately before an ICA injury are valuable for transsphenoidal/pituitary surgeons to avoid this catastrophic incident. Furthermore, operative video digests from immediately before ICA injury to completion of hemostasis are extremely valuable. The authors are expected to provide the operative video digests. 

Respond#:  Thanks for pointing out this issue. We have recently added an intraoperative video, including immediate before ICA injury, during the damage, and completion of injury scenes. We have added an explanation to the last paragraph of the material method section: ”An intraoperative video of case 7 in table 1 demonstrates the ICA injury during the endonasal endoscopic opening of the cavernous sinus's medial wall and tight packing hemostatic material with successful hemostasis. The video also indicates the occurrence of a pseudoaneurysm in the injured site and its treatment with endovascular micro coils in the postoperative angiography (Video 1). “

Reviewer 2 Report

English language is not good enough and should be corrected.

There are no letter indicators in figures 2, 3, 4, and 5, although these are explained in agenda. The numbers are inverted in figure 5B.

Author contributions are not provided.

Conclusions are drawn not from the processes, that are discussed in the text. The main questions that are being discussed in the text are the defects of ICA and the ways of correction. However the main conclusion is recommendation to study anatomy before the surgery. Consider revision of the conclusion section.

Author Response

Reviewer 2

Respond#: We would like to thank the reviewer for his/her many helpful suggestions.

Reviewer:  English language is not good enough and should be corrected.

Respond#: We have gone through the manuscript to correct some syntax errors. Please check the latest version of the text.

Reviewer:  There are no letter indicators in figures 2, 3, 4, and 5, although these are explained in agenda. The numbers are inverted in figure 5B.

Respond#:  Thanks for pointing it out! We have recently added new arrows and revised the figure legends of Figures 2, 3, and 4. Also, we have corrected Figure 5B. Additionally, we have edited Figure 1.

Reviewer:  Author contributions are not provided.

Respond#: We have recently added author contributions.

Reviewer:  Conclusions are drawn not from the processes, that are discussed in the text. The main questions that are being discussed in the text are the defects of ICA and the ways of correction. However the main conclusion is recommendation to study anatomy before the surgery. Consider revision of the conclusion section.

Respond#: We have recently changed the conclusion part briefly, indicating the importance of the tools used to avoid ICA injury before and during the surgery and solutions to fix the ICA injury during and after surgery. Please see the edited version of the conclusion as follows: Meticulous neuroimaging data analysis and surgeon’s experience can significantly reduce the risk of iatrogenic ICA injury during the transsphenoidal approach. When removing the pituitary gland and parasellar region's tumors using the endoscopic transsphenoidal access, correct identification of the midline is of paramount importance. To prevent complications, it is advisable to use intraoperative navigation and ultrasound dopplerography to visualize the ICA during procedures performed in the parasellar region. Dealing with the ICA injury requires detailed angiographic examination and a combined multidisciplinary treatment approach.”